# Differences in Accuracy and Radiation Dose in Placement of Iliosacral Screws: Comparison between 3D and 2D Fluoroscopy

**DOI:** 10.3390/jcm11061466

**Published:** 2022-03-08

**Authors:** Michał Kułakowski, Paweł Reichert, Karol Elster, Jarosław Witkowski, Paweł Ślęczka, Piotr Morasiewicz, Łukasz Oleksy, Aleksandra Królikowska

**Affiliations:** 1Independent Public Healthcare Centre, 87-500 Rypin, Poland; mkulakowski@poczta.fm (M.K.); karol.elster@gmail.com (K.E.); 2Clinical Department of Trauma and Hand Surgery, Department of Trauma Surgery, Faculty of Medicine, Wroclaw Medical University, 50-556 Wrocław, Poland; 3Clinical Department of Trauma and Hand Surgery, Division of Sports Medicine, Department of Trauma Surgery, Faculty of Medicine, Wroclaw Medical University, 50-556 Wrocław, Poland; jaroslaw.witkowski@umw.edu.pl; 4Independent Public Healthcare Centre, 32-400 Myślenice, Poland; pawelsleczka@wp.pl; 5Department of Orthopaedic and Trauma Surgery, University Hospital in Opole, Institute of Medical Sciences, University of Opole, 45-401 Opole, Poland; piotr.morasiewicz@uni.opole.pl; 6Orthopaedic and Rehabilitation Department, Medical Faculty, Medical University of Warsaw, 02-091 Warsaw, Poland; loleksy@oleksy-fizjoterapia.pl; 7Physiotherapy and Sports Centre, Rzeszow University of Technology, 35-959 Rzeszow, Poland; 8Ergonomics and Biomedical Monitoring Laboratory, Department of Physiotherapy, Faculty of Health Sciences, Wroclaw Medical University, 50-367 Wrocław, Poland; aleksandra.krolikowska@umw.edu.pl

**Keywords:** iliosacral screw, 3D fluoroscopy, 2D fluoroscopy, pelvic ring injuries, percutaneous fixation

## Abstract

Percutaneous iliosacral screw fixation is a widely accepted method of stabilizing the posterior pelvic ring. Recently developed tools such as 3D-navigated fluoroscopy and computed navigation seem to prevent a surgeon from conducting screw misplacement. The study aimed to comparatively assess the introduction of sacroiliac screw placement using 2D and 3D fluoroscopy in terms of accuracy and radiation exposure. Iliosacral screws were introduced in 37 patients using 2D (group N1) and in 36 patients using 3D fluoroscopy (group N2) techniques. Overall, 61 and 56 screws were introduced in groups N1 and N2, respectively. Screw placement accuracy was assessed using postoperative computed tomography and Smith’s scale. Intraoperative radiation exposure was also assessed. No differences were noted between groups in terms of screw positioning accuracy and radiation dose. Both 2D and 3D fluoroscopy provide good visualization for safely placing percutaneous iliosacral joint screws. Using 3D fluoroscopy-based navigation in comparison with 2D fluoroscopy is not advantageous.

## 1. Introduction

Pelvic fractures show two peak incidences in general. The first peak is around 18–35 years and most cases result from high-energy trauma. The second peak is observed in the female population over 70 years old and is strictly related to bone quality [1]. The mortality varies from 7.9% to 10% when ‘the complex pelvic trauma’ definition is fulfilled. We are dealing with a 20% mortality rate when unstable circulation is added [2].

Iliosacral percutaneous screw fixation is a frequently used, minimally invasive method of treatment of posterior pelvic ring injuries [3]. Indications for this technique include iliosacral joint dislocation, sacral fractures, and crescent fractures. Iliosacral screws have multiple advantages, including good biomechanical stability [1]. The use of intraosseous fixation avoids surface contamination as well as implant prominence and skin breakdown [4].

In the past, the open technique was applied with direct visualization of the injury and implant [5]. This method was associated with a high risk of complications. Matta and Saucedo introduced minimally invasive percutaneous screw placement [6]. This technique, which was pioneered in a prone position, was later modified, allowing its use in a supine position with or without the percutaneous approach [3]. The use of the percutaneous method avoids the high rate of wound necrosis and infection associated with open procedures [7]. Despite its widespread acceptance and use, iliosacral screw placement is challenging for various reasons (variable posterior ring anatomy and upper sacral segment dysmorphism).

To achieve reduction and proper screw fixation, many authors use fluoroscopy, computed tomography, and computer-assisted methods. Some of them use two (inlet and outlet) or three (inlet, outlet, and lateral) fluoroscopic views [8].

In many publications, high misplacement rates are observed, but navigated procedures were introduced to decrease the radiation and increase the accuracy of screw placement, especially in percutaneous techniques [9].

The role of 3D intraoperative imaging in orthopaedic trauma is presented in a paper by Tonetti et al. [10].

Screw malposition rates using fluoroscopy vary from 2% to 15%, but some authors notice that using computer-assisted techniques creates a higher rate of complications [11,12]. Neurovascular complication rates vary from 0.5% to 7% [13]. It is not clear in the literature if using advanced visualization methods decreases complication rates. Therefore, the authors decided to conduct this study and determine whether using 3D fluoroscopy in comparison with 2D fluoroscopy is favourable and whether it is necessary to invest in expensive, sophisticated visualization methods.

The aim of this study was to compare the accuracy of screw placement and radiation exposures using 2D and 3D fluoroscopy.

## 2. Materials and Methods

### 2.1. Population

The study included 73 patients aged 19–75 years (mean 41.96, standard deviation ± 14.69, median 42) treated surgically by closed reduction and internal fixation after sacral injury. Patients were operated on using a standard technique with 2D X-Ray (N1 group = 37) or with 3D CT scan (N2 group = 36).

A retrospective comparison of those two techniques in terms of the correct position of fixating screws and radiation dose was performed.

The study was approved by the local bioethics committee.

All participants signed informed consent forms to use their anonymous medical data for research purposes.

From 2015 to 2020, 73 patients were treated in the orthopaedic and trauma departments in City Hospital in Torun and the Independent Public Healthcare Centre in Rypin, Poland. All patients presented sacral fractures—47 in zone 1, according to the Denis Classification, and 26 in zone 2. Patients with sacroiliac joint dislocations were excluded from the study. All 73 patients underwent percutaneous iliosacral screwing by one experienced surgeon. Before beginning this study, 32 screws were inserted by the first author.

All procedures were performed in a supine position.

In the case of 37 patients, they were operated on using 2D fluoroscopy (Ziehm Vision R, Nuremberg, Germany), and 36 were operated on using 3D fluoroscopy (Ziehm Vision FD Vario 3D, Nuremberg, Germany). The availability of C-arm was random.

In the 2D group, 20 patients were stabilized with 2 screws, and 17 with 1 screw. In 14 patients, 1 screw was inserted in S1, and in 3 cases, S1 dysmorphism was recognized and these patients were fixed with 1 screw in S2.

In the 3D group, 20 patients were stabilized with 2 screws and 16 with one screw, 13 in S1 and 3 in S2.

### 2.2. Technique Procedures

All patients were operated on in a supine position with general anaesthesia on a radiolucent table. A total of 53 patients initially underwent stabilization and reduction of the anterior pelvic ring, 41 with plates and 12 with intramedullary screws. In 20 cases, only posterior stabilisation was performed.

We used the entry point in supine position, which was 2 cm cranially from the ASIS, confirmed by the lateral view recommended by Giannoudis et al. [14].

Having confirmed the entry point, a skin incision was made, and a K-wire of 1.8 mm was inserted just medially from the sacroiliac joint. After that, inlet and outlet views were made and if the position of the wire was correct, it was inserted to the midline of the sacral bone and a 6.5 mm cannulated screw (DePuy Synthes) was inserted. The second screw was inserted with the aid of an aiming guide, also under fluoroscopic vision.

Another round of inlet and outlet views were made.

In the 3D group, K-wires were inserted using 2D checks and, having achieved their final positions (in the middle of the sacrum), a 3D scan was made. Another 3D scan was made when the screw was inserted. If we decided to put the second screw, insertion of the K-wire was performed with the use of aiming guide and using 2D checks and 3D scan was made also when the final, correct position of the pin was achieved. Having inserted the second screw, another 3D scan was performed.

The information about the emitted radiation dose was read from C-arms.

To assess the positioning of the screws, postoperative CT scans were made.

### 2.3. Statistical Analysis

The results were statistically analysed using Statistica 13.3 software (TIBCO Software Inc., Palo Alto, CA, USA). Arithmetic means and standard deviations (±) were calculated. The Shapiro–Wilk test was used to analyse the compliance of the distribution of the analysed variables with the normal distribution.

The Student’s *t*-distribution test was used to evaluate the significance of differences between patients’ ages in the two groups. For all data without normal distribution, the Mann–Whitney U test was performed. Spearman’s rank correlation coefficient was used to confirm the relationship between the X-ray dose and other variables.

Statistical significance was established at the level of *p* ≤ 0.050.

## 3. Results

Early postoperative CT scans were made to evaluate the positions of the screws. The most important factor was to identify cortical and neuroforaminal perforations. To assess the perforation, we used the scale suggested by Smith et al. [15]: grade 0, no perforation (Figure 1); grade 1, perforation less than 2 mm (Figure 2); grade 2, perforation between 2 and 4 mm (Figure 3); and grade 3, perforation more than 4 mm (in group 3, insertion of a screw in a foramen was also included).

### 3.1. Radiation Dose

There was no significant difference (*p* = 0.46) between radiation doses in N1 (20.47 ± 6.0, median 18.6 mGy) and N2 (19.55 ± 6.16, median 17.3 mGy). The main differences in the radiation doses depending on the number of screws in groups N1 and N2 are shown in the table below (Figure 4).

There was no significant Spearman correlation between radiation doses and patients’ ages in both groups together (*p* = 0.37) and separately for N1 (*p* = 0.85) and N2 (*p* = 0.18). No significant correlation between radiation doses and numbers of screws (*p* = 0.49, N1 *p* = 0.73, N2 *p* = 0.2) or incorrectly placed screws (*p* = 0.39, N1 *p* = 0.69, N2 *p* = 0.13) was found.

### 3.2. Number, Localization, and Correct Position of Fixation Screws

There was no significant difference (*p* = 0.95) between the number of fixation screws used for stabilisation in groups N1 (1.56 ± 0.5) and N2 (1.56 ± 0.5). Only one screw was used in 43.59% and 44.4% cases from Group N1 and N2, respectively.

The localisation of fixation screws was similar in both groups. The screw placed in S1 was in 92.31% and 91.7% cases from Group N1 and N2, respectively. The screw placed in S2 was in 64.1% and 63.89% cases from Group N1 and N2, respectively. Those differences were statistically insignificant with *p* = 0.97 and *p* = 0.99, respectively.

The number of incorrectly placed screws was similar between both groups (*p* = 0.77) and these constituted 11.48% and 8.93% of all screws for N1 and N2, respectively. The absolute number of patients with at least one screw incorrectly placed was seven (17.95%) in group N1 and five (13.89%) in group N2. The most incorrectly placed screws in the N1 group (5 cases) were with sacral bone perforations of less than 2 mm (stage I according to Smith’s scale) and two cases were with bone perforations of more than 4 mm (stage III). In group N2, 3 of 5 incorrectly placed screws were estimated as stage I, and 2 as stage II (sacral bone perforation between 2 and 4 mm) (Figure 5).

### 3.3. Operative Time

In the navigation group, the operative time was significantly longer (25.72 min. versus 18.61 min., *p* < 0.001).

## 4. Discussion

In this study, we compared fluoroscopically guided iliosacral screw placement under 2D and 3D C-arm. All procedures were performed by the first author, an experienced surgeon according to criteria suggested by Verbeek et al. [16]. In their study, 11 surgeons were divided into three groups: experts performed over 25 screws, experienced surgeons over 10, and less experienced surgeons below 10 screws. He could observe a higher malposition rate among less experienced surgeons, and further, a tendency toward a lower failure rate was observed with higher surgical experience. A recent study also pointed out that malposition rates are influenced by surgeons’ experience [17].

### 4.1. Visualization

We found it extremely useful to perform the lateral view first, as confirmed by former studies, followed by inlet and outlet views [14]. Using the lateral view, we assessed the lateral sacral triangle as a decision support for secure screw insertion in S1 [18]. New reports suggesting the correlating preoperative imaging with intraoperative fluoroscopy seem to present a new approach to preoperative planning in iliosacral screw placement [19]. In our series, we found sacral dysmorphism in 8.3% of cases, although it has been reported to be identified in almost half of the adult population. The size of the sacral triangle and acute alar slope as described by Miller and Routt led us to the recognition of sacral dysmorphism and, in these cases, we inserted screws only in S2 [20,21]. This approach is similar to that presented by Moed [22].

### 4.2. Screw Positioning

In 2013, a systematic review on iliosacral screw fixation compared the accuracy and revision rates for different imaging modalities [23]. Despite a lack of clear definition of screw malposition and a great variety of study outcomes, the authors revealed the superiority of CT navigation to 2D fluoroscopy, but not to computer-navigated surgery. In our study, we did not identify the superiority of any visualisation method. Further, Verbeek et al., in their study from 2016, did not demonstrate the superiority of 3D navigation to conventional fluoroscopy [16]. 

Screw malposition rates with fluoroscopic guidance have been reported as being from 2% to 15% [9,11]. The incidence of neurologic injury is reported as between 0.5% and 7.7%. [24]. Zwingmann et al. revealed more neurologic complications among navigation groups (14%) than fluoroscopically assisted screw insertion (2.1%) [23]. In this study, only one screw was inserted in the sacral S1 foramen and the patient suffered from transient neurologic deficits, but refused screw removal. The malposition rate is also associated with a higher number of inserted screws and the insertion of a screw in S2 [25]. The number of screws required for pelvic ring injury fixation remains controversial. Ebrahim et al. suggested the insertion of one screw in S1, whereas others recommended the placement of two screws [26,27]. This approach is similar to that presented by Rommens [28].

### 4.3. Radiation

We achieved differences in radiation dose, but not statistically significant ones. Further, Verbeek et al. did not report any difference in radiation dose but Zwingmann noticed a reduction in the radiation dose in the navigation group [16,23]. This difference is explained by higher radiation doses in the fluoroscopic group that is associated with difficulties in the correct positioning of the C-arm, which is not needed in a navigation system. A 3D C-arm is about three times more expensive than a 2D C-arm.

Theologis et al. found significantly higher radiation among the O-Arm group than the fluoroscopy group [29]. Similar findings are presented by Boudissa [8].

LIMITATIONS: The radiation dose collection using reports from the device (C-arm, O-arm) is not a very precise method of determining a patient’s exposure to radiation, and this may be considered a limitation. The radiation dose is also dependent on the BMI of the patient.

We did not assess long results of our treatment and the influence of using one or two screws on bone healing. Further, the number of screws required for stable fixation remains unclear. Another limitation of this study may be the relatively small number of patients.

### 4.4. Operative Time

In our study, the operative time was higher in the 3D group, which is similar to the results presented by Boudissa et al. [8]. However, some authors reported decreased operative time when using navigation [29].

### 4.5. Power Calculation

We performed power calculation of our statistics. The power of screw positioning and operative time tests was over 0.8. The smallest power was found in radiation dose and was calculated as 0.64. Moreover, the estimated number of patients needed to detect a robust statistical difference was 102 in each group.

## 5. Conclusions

Both 2D and 3D fluoroscopy provide good visualization for safely placing percutaneous iliosacral joint screws. Using 3D fluoroscopy-based navigation in comparison with 2D fluoroscopy is not advantageous.

## Figures and Tables

**Figure 1 jcm-11-01466-f001:**
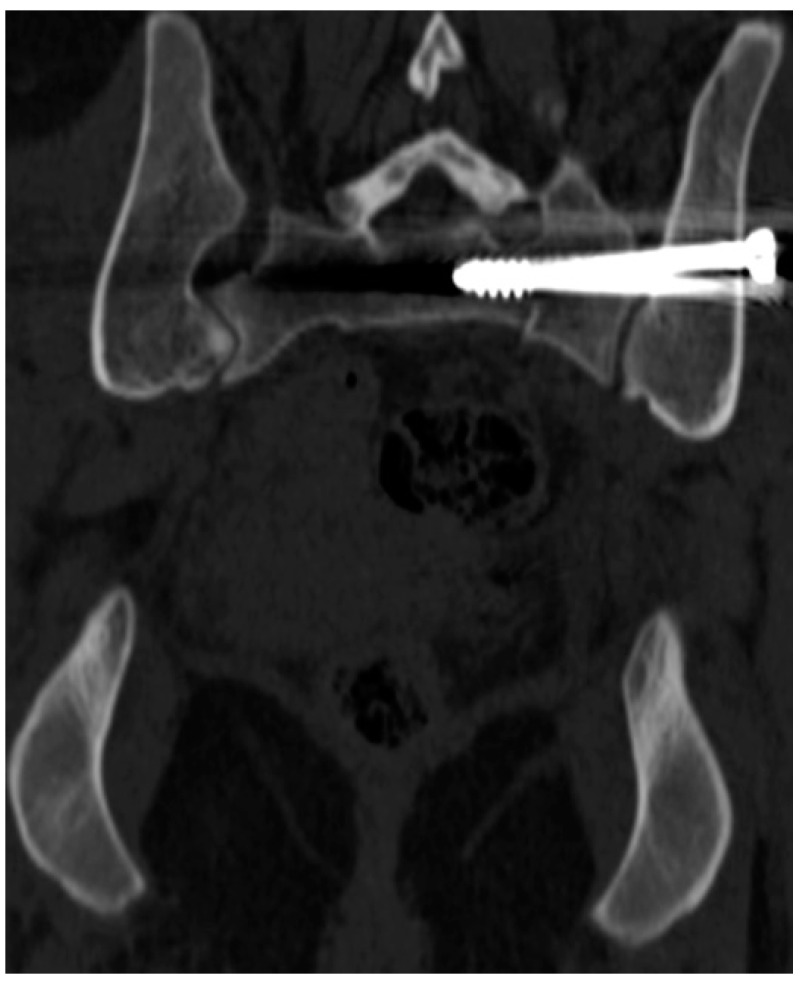
Screw inside the sacrum (Smith type 0).

**Figure 2 jcm-11-01466-f002:**
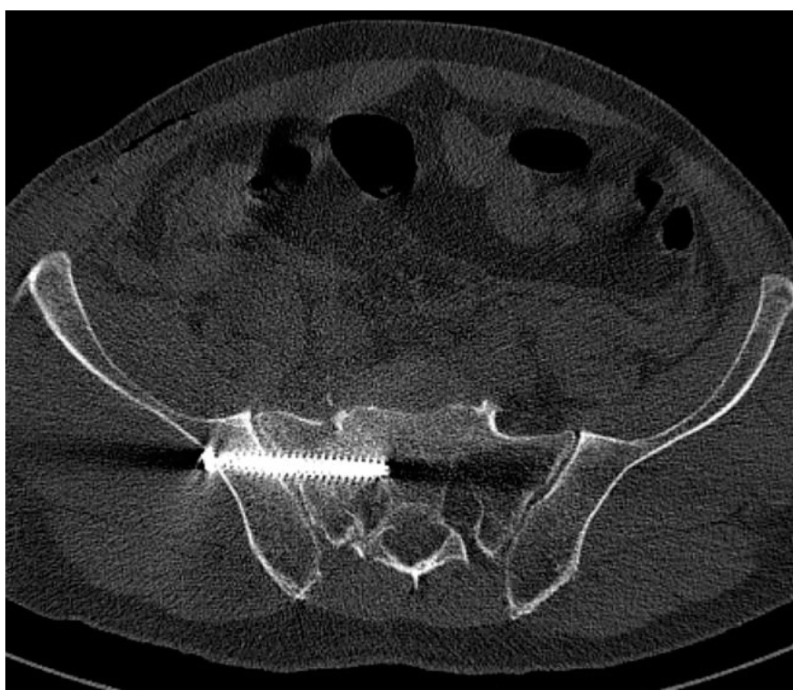
Screw touching the cortex (Smith type 1).

**Figure 3 jcm-11-01466-f003:**
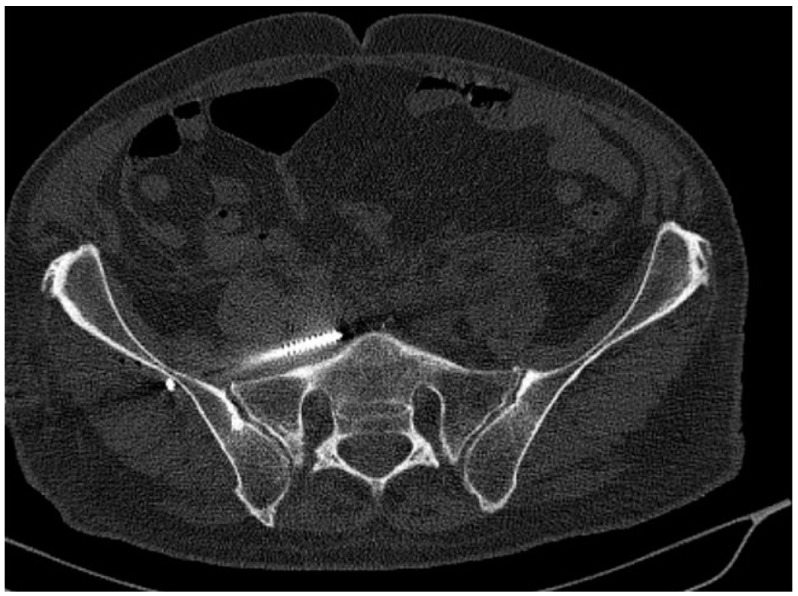
Screw anterior from S1—unrecognised sacral dysmorphism (Smith type-3 misplacement).

**Figure 4 jcm-11-01466-f004:**
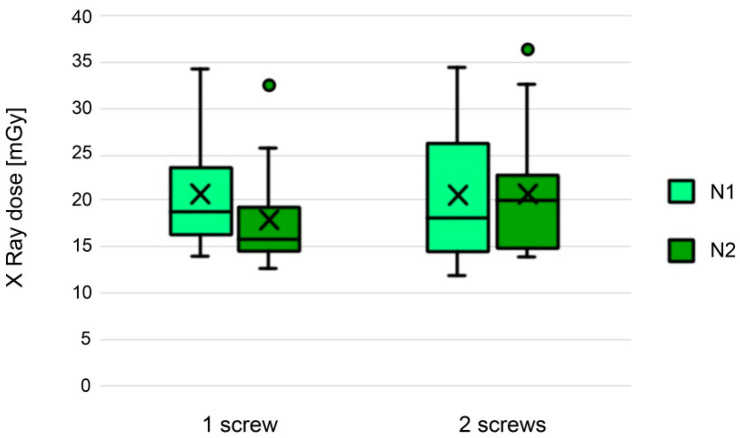
Box-and-whisker plot showing differences in the radiation doses depending on the number of screws in group N1 and N2. (x—mean; line—median; box—first to third quartile; whiskers—minimum and maximum; dot—data outliers).

**Figure 5 jcm-11-01466-f005:**
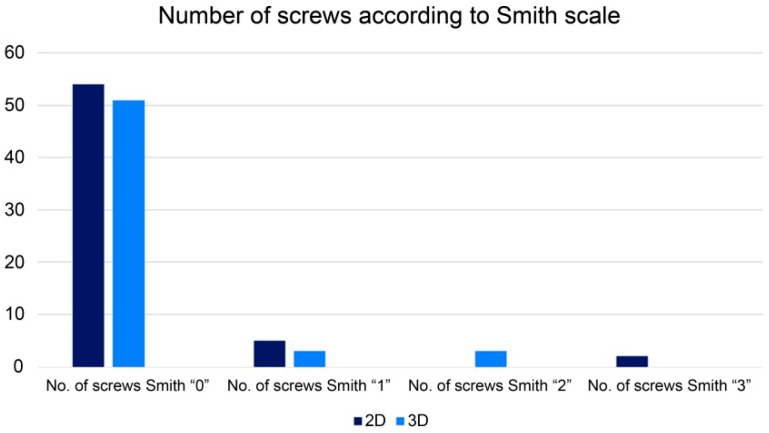
Number of screws in Group N1 (2D) and N2 (3D) according to the Smith scale.

## Data Availability

Not applicable.

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
