# Peer review of "Differences in Accuracy and Radiation Dose in Placement of Iliosacral Screws: Comparison between 3D and 2D Fluoroscopy"

_jcm, 2022, doi:10.3390/jcm11061466_

Round 1

Reviewer 1 Report

Ce papier va à contre courant des publications sur la navigation chirurgicale en traumatologie.

[Boudissa M, Carmagnac D, Kerschbaumer G, Ruatti S, Tonetti J. Screw Misplacement in Percutaneous Posterior Pelvic Iliosacral Screwing with and without Navigation: A Prospective Clinical Study of 174 Screws in 127 Patients. Orthop Traumatol Surg Res. 2022 Jan 23:103213. doi: 10.1016/j.otsr.2022.103213.] et

Ses conclusions doivent donc être étayées sur des preuves scientifiques solides, méthodologiques en particulier.

Dans le chapître méthode, les mesures de doses dépendent du volume du patient . Nous devons savoir quels types de capteurs on été utilisés. S'il s'agit uniquement des informations de la machine la dose réelle n'est pas obtenue en lecture directe mais après correction  avec la corpulence du patient. L'imagerie 3D peut être comparées en utilisant le produit dose.surface (PDS) (cGray.cm2).[Tonetti J, Boudissa M, Kerschbaumer G, Seurat O. Rôle de l’imagerie peropératoire 3D en chirurgie orthopédique et traumatologique. Orthop Traumatol Surg Res. 2020 Fév;106(1S):S19-S25. doi: 10.1016/j.otsr.2019.05.021.]

Le dispositif de navigation 3D Ziehm n'est pas décrit. On ne sais pas si l'oérateur navigue dans un volume de Voxel, s'il fait des contrôle 2D à a progression de la broche et de l'implant, s'il fait un contrôle 3D post-opératoire.

La durée opératoire par vis et de chaque procédure doit être connue.

Le fait d'avoir un opérateur très entrainé ne permets pas d'éprouver la robustesse des 2 méthodes. 

Les conclusions sont radicales sans débat de la littérature récente. Cet article est trop le témoin de la pratique personnelle d'un seul opérateur qui n'éprouve pas le besoin de faire l'effort d'avoir un outil de navigation.

Base scientifiques insuffisantes 

Author Response

17th February 2022

First of all I would like to thank for constructive and substantive comments. We believe that the manuscript gained a lot after the corrections. Below, there are included individual comments of the reviewers, as well as the responses and changes introduced.

  • What kind of sensors were used. If it is only information from the machine?

We used the information from the machine. We know that emited radiation is different from absorbed by patients, but for the comparative purpose, information from the machine seems to be sufficient.

  • Ziehm 3D navigation is not described”

We used Ziehm Vision FD Vario 3D

  • We do not know if the operator navigates in Voxel volume, if he does 2D checks on the progress of the pin and the implant, if he does a post-operative 3D checks?

2.2 Technique procedures, line 115

  1. Previous version:

In the 3D group, 3D scans were made.

  1. b) Current version:

In the 3D group K-wires were inserted using 2D check and having achieved final position (in the middle of the sacrum) a 3D scan was made. Another 3D scan was made when the screw was inserted. If we decided to put the second screw, insertion of the K-wire was performed with the use of aiming guide and using 2D checks and 3D scan was made also when final, correct position of the pin was achieved. Having inserted the second screw, another 3D scan was performed.

  • The operating time per screw and each procedure must be known.

We added point 3.4 in results and 4.4 in discussion:

Current version:

3.4 Operative time

In the navigation group the operative time was significantly longer (25,72 min. versus 18,61 min., p<0,001)

4.4 Operative time

In our study the operative time was higher in 3-D group that is similar to results presented by Boudissa et al. [8]. Although some authors reported decreased operative time when using navigation  [29].

  • Comment: The fact of having a very trained operator does not allow testing the robustness of the 2 methods.

The study was performed by one person, a trained pelvic surgeon, that can minimize potential bias related to learning curve and different surgeons.

According to reviewers comments spelling mistakes were checked again and corrected.

All old references (despite classic ones) were removed and changed with new ones, as follows:

Removed references:

  1. Comstock, C.P.; van der Meulen, M.C.; Goodman, S.B. Biomechanical comparison of posterior internal fixation techniques for unstable pelvic fractures. J Orthop Trauma 1996,10(8),517-522.
  2. Pohlemann, T.; Tosounidis, G.; Bircher, M.; Giannoudis, P.; Culemann U. The German Multicentre Pelvis Registry: a template for an European Expert Net-work? Injury 2007,38(4),416-423. doi: 10.1016/j.injury.2007.01.007.
  3. Moed, B.R.; Whiting, D.R. Locked transsacral screw fixation of bilateral injuries of the posterior pelvic ring: Initial Clinical Series. J Orthop Trauma 2010,24(10),616-621.
  4. Shuler, T.E.; Boone, D.C.; Gruen, G.S. Percutaneous iliosacral screw fixation: early treatment for unstable posteriori pelvic ring disruptions. J Trauma 1995,38,453-458.
  5. Schep, N.W.; Haverlag, R.; van Vugt, A.B. Computer assisted versus conventional surgery for insertion of 96 cannulated iliosacral screws in patients with postpartum pelvic pain. J Trauma 2004,57,1299-1302.
  6. Briem, D.; Rueger, J.M.; Begemann, P.G. Computer-assisted screw placement in-to the posterior pelvic ring: assessment of different navigated procedures in a cadaver trial. Unfallchirurg 2006,109,640-646.
  7. Templeman, D.; Schmidt, A.; Freese, J.; Weisman, I. Proximity of iliosacral screws to neurovascular structures after internal fixation. Clin Orthop Retal Res 1996,329,194-198.
  8. Hinsche, A.F.; Giannoudis, P.V.; Smith, R.M. Fluoroscopy based multiplanar image guidance for insertion of iliosacral screws. Clin Orthop Relat Res 2002,395,135-144.
  9. Collinge, C.; Coons, D.; Tornetta, P.; Aschenbrenner, J. Standatrd multiplanar fluoroscopy versus a fluoroscopically based navigation system for the percutaneous insertion of iliosacral screws: a cadaver model. J Orthop Trauma 2005,19,254-258.
  10. Grossterlinden, L.; Rueger, J.; Catala-Lehnen, P.; Rupprecht, M.; Lehmann, W.; Rücker, A.; Briem, D. Factors influencing the accuracy of iliosacral screw placement in trauma patients. Int Orthop 2011,35,1391-1396.

And changed with

  1. Melhem E, Riouallon G, Habboubi K, Gabbas M, Jouffroy P.Epidemiology of pelvic and acetabular fractures in France. Orthop Traumatol Surg Res. 2020 Sep;106(5):831-839.
  2. Bakhshayesh P, Weidenhielm L, Enocson A.Factors affecting mortality and reoperations in high-energy pelvic fractures. Eur J Orthop Surg Traumatol. 2018 Oct;28(7):1273-1282.
  3. Wenning KE, Yilmaz E, Schildhauer TA, Hoffmann MF.Comparison of lumbopelvic fixation and iliosacral screw fixation for the treatment of bilateral sacral fractures. J Orthop Surg Res. 2021 Oct 16;16(1):604.
  4. Jaeblon T, Perry KJ, Kufera JA.Waist-Hip Ratio Surrogate Is More Predictive Than Body Mass Index of Wound Complications After Pelvic and Acetabulum Surgery. J Orthop Trauma. 2018 Apr;32(4):167-173.
  5. Boudissa M, Carmagnac D, Kerschbaumer G, Ruatti S, Tonetti J. Screw Misplacement in Percutaneous Posterior Pelvic Iliosacral Screwing with and without Navigation: A Prospective Clinical Study of 174 Screws in 127 Patients. Orthop Traumatol Surg Res. 2022 Jan 23:103213.
  6. San Miguel-Ruiz JE, Polly D, Albersheim M, Sembrano J, Takahashi T, Lender P, Martin CT.Is the Implant in Bone? The Accuracy of CT and Fluoroscopic Imaging for Detecting Malpositioned Pelvic Screw and SI Fusion Implants. Iowa Orthop J. 2021;41(1):89-94.
  7. Quade J, Busel G, Beebe M, Auston D, Shah AR, Infante A, Maxson B, Watson D, Sanders RW, Mir HR.Symptomatic Iliosacral Screw Removal After Pelvic Trauma-Incidence and Clinical Impact. J Orthop Trauma. 2019 Jul;33(7):351-353.
  8. Berger-Groch J, Lueers M, Rueger JM, Lehmann W, Thiesen D, Kolb JP, Hartel MJ, Grossterlinden LG.Accuracy of navigated and conventional iliosacral screw placement in B- and C-type pelvic ring fractures. Eur J Trauma Emerg Surg. 2020 Feb;46(1):107-113.
  9. Maslow J, Collinge CA. Risks to the Superior Gluteal Neurovascular Bundle During Iliosacral and Transsacral Screw Fixation: A Computed Tomogram Arteriography Study. J Orthop Trauma. 2017 Dec;31(12):640-643.
  10. Araiza ET, Medda S, Plate JF, Marquez-Lara A, Trammell AP, Aran FS, Lara D, Danelson K, Halvorson JJ, Carroll EA, Pilson HT.Comparing the Efficiency, Radiation Exposure, and Accuracy Using C-Arm versus O-Arm With 3D Navigation in Placement of Transiliac-Transsacral and Iliosacral Screws: A Cadaveric Study Evaluating an Early Career Surgeon.

J Orthop Trauma. 2020 Jun;34(6):302-306.

We also added following references:

  1. Boudissa M, Carmagnac D, Kerschbaumer G, Ruatti S, Tonetti J. Screw Misplacement in Percutaneous Posterior Pelvic Iliosacral Screwing with and without Navigation: A Prospective Clinical Study of 174 Screws in 127 Patients. Orthop Traumatol Surg Res. 2022 Jan 23:103213.
  2. Tonetti J, Boudissa M, Kerschbaumer G, Seurat O. Role of 3D intraoperative imaging in orthopedic and trauma surgery. Orthop Traumatol Surg Res (2019);106(1S):S19-S25.
  3. Theologis AA, Burch S, Pekmezci M. Placement of iliosacral screw using 3D image-guided (O-Arm) technology and Stealth Navigation. Bone Joint J 2016;98-B:696-702
  4. Rommens PM, Nolte EM, Hopf J, Wagner D, Hofmann A, Hessmann M.Safety and efficacy of 2D-fluoroscopy-based iliosacral screw osteosynthesis: results of a retrospective monocentric study. Eur J Trauma Emerg Surg. 2021 Dec;47(6):1687-1698.
  5. Eastman JG, Routt Jr M. Correlating preoperative imaging with intraoperative fluoroscopy in iliosacral screw placement. J Orthopaed Traumatol 2015,16:309-316

We do hope that so far, we have improved the manuscript sufficiently to the major remarks. We are very glad for the reviewer's valuable comments.

Reviewer 2 Report

Good paper. Simple and reads quite easily. A few spelling mistakes identified. these need to be changed. References however, are old. 22 of 25 references are older than 10 years old. Please do a literature review and include some newer references. Discard old references unless they are classics.

Just a few suggestions. 1) Eastman and Routt 2015 J of Orthopedics and traumatology. 2) Theologis and Burch and Pekmezci 2016 BJJ 98(5). 3) Rommens and Nolte and Hopf European J of Trauma and Emerg Surg 2021 47(6).

Intro - Line 42 - This is not at all clear. What do you mean 20%. Rewrite and make clear.

         - Line 66-70 - This is 1 sentence and needs to be at least 2 sentences. Rewrite.

         - Check spelling in Intro as this reviewer saw some spelling errors.

M and M - Line 97-98 - 36 patients into 20 (2) and 16(1) but 10 and 3 do not add up. Clarify please.

Results - Line 119 - "foramed" - do you mean "foramen"?

Author Response

17th February 2022

First of all I would like to thank for constructive and substantive comments. We believe that the manuscript gained a lot after the corrections. Below, there are included individual comments of the reviewers, as well as the responses and changes introduced.

  • Comment nr 1- line 42: This is not clear. What do you mean 20%. Rewrite and make clear.
  1. Previous version:

We are dealing with 20%  when unstable circulation is added.

  1. Current version :

We are dealing with 20% mortality rate when unstable circulation is added.

  • Comment nr 2 – line 66-70: This is one sentence and needs to be at least 2 sentences. Rewrite.
  1. Previous version:

Due to different results in papers using different visualisation methods, that is, 2D or 3D fluoroscopy or computed navigation, the authors decided to conduct this study and determine whether using 3D fluoroscopy in comparison with 2D is favourable and whether it is necessary to invest in expensive, sophisticated visualisation methods.

  1. Current version:

It is not clear in the literature if using advanced visualization methods decreases complication rates. Therefore the authors decided to conduct this study and determine whether using 3D fluoroscopy in comparison with 2D is favourable and whether it is necessary to invest in expensive, sophisticated visualisation methods.

  • Comment nr 3 – line 97-98: 36 patients into 20(2) and 16(1) but 10 and 3 do not up. Clarify please.
  1. Previous version:

In the 3D group, 20 patients were stabilised with 2 screws and 16 with one screw, 10  in S1 and 3 in S2.

  1. Current version:

In the 3D group, 20 patients were stabilised with 2 screws and 16 with one screw, 13 in S1 and 3 in S2.

  • Comment nr 4 – Line 119

  1. Previous version

Foramed

  1. Current version

Foramen

According to reviewers comments spelling mistakes were checked again and corrected.

All old references (despite classic ones) were removed and changed with new ones, as follows:

Removed references:

  1. Comstock, C.P.; van der Meulen, M.C.; Goodman, S.B. Biomechanical comparison of posterior internal fixation techniques for unstable pelvic fractures. J Orthop Trauma 1996,10(8),517-522.
  2. Pohlemann, T.; Tosounidis, G.; Bircher, M.; Giannoudis, P.; Culemann U. The German Multicentre Pelvis Registry: a template for an European Expert Net-work? Injury 2007,38(4),416-423. doi: 10.1016/j.injury.2007.01.007.
  3. Moed, B.R.; Whiting, D.R. Locked transsacral screw fixation of bilateral injuries of the posterior pelvic ring: Initial Clinical Series. J Orthop Trauma 2010,24(10),616-621.
  4. Shuler, T.E.; Boone, D.C.; Gruen, G.S. Percutaneous iliosacral screw fixation: early treatment for unstable posteriori pelvic ring disruptions. J Trauma 1995,38,453-458.
  5. Schep, N.W.; Haverlag, R.; van Vugt, A.B. Computer assisted versus conventional surgery for insertion of 96 cannulated iliosacral screws in patients with postpartum pelvic pain. J Trauma 2004,57,1299-1302.
  6. Briem, D.; Rueger, J.M.; Begemann, P.G. Computer-assisted screw placement in-to the posterior pelvic ring: assessment of different navigated procedures in a cadaver trial. Unfallchirurg 2006,109,640-646.
  7. Templeman, D.; Schmidt, A.; Freese, J.; Weisman, I. Proximity of iliosacral screws to neurovascular structures after internal fixation. Clin Orthop Retal Res 1996,329,194-198.
  8. Hinsche, A.F.; Giannoudis, P.V.; Smith, R.M. Fluoroscopy based multiplanar image guidance for insertion of iliosacral screws. Clin Orthop Relat Res 2002,395,135-144.
  9. Collinge, C.; Coons, D.; Tornetta, P.; Aschenbrenner, J. Standatrd multiplanar fluoroscopy versus a fluoroscopically based navigation system for the percutaneous insertion of iliosacral screws: a cadaver model. J Orthop Trauma 2005,19,254-258.
  10. Grossterlinden, L.; Rueger, J.; Catala-Lehnen, P.; Rupprecht, M.; Lehmann, W.; Rücker, A.; Briem, D. Factors influencing the accuracy of iliosacral screw placement in trauma patients. Int Orthop 2011,35,1391-1396.

And changed with

  1. Melhem E, Riouallon G, Habboubi K, Gabbas M, Jouffroy P.Epidemiology of pelvic and acetabular fractures in France. Orthop Traumatol Surg Res. 2020 Sep;106(5):831-839.
  2. Bakhshayesh P, Weidenhielm L, Enocson A.Factors affecting mortality and reoperations in high-energy pelvic fractures. Eur J Orthop Surg Traumatol. 2018 Oct;28(7):1273-1282.
  3. Wenning KE, Yilmaz E, Schildhauer TA, Hoffmann MF.Comparison of lumbopelvic fixation and iliosacral screw fixation for the treatment of bilateral sacral fractures. J Orthop Surg Res. 2021 Oct 16;16(1):604.
  4. Jaeblon T, Perry KJ, Kufera JA.Waist-Hip Ratio Surrogate Is More Predictive Than Body Mass Index of Wound Complications After Pelvic and Acetabulum Surgery. J Orthop Trauma. 2018 Apr;32(4):167-173.
  5. Boudissa M, Carmagnac D, Kerschbaumer G, Ruatti S, Tonetti J. Screw Misplacement in Percutaneous Posterior Pelvic Iliosacral Screwing with and without Navigation: A Prospective Clinical Study of 174 Screws in 127 Patients. Orthop Traumatol Surg Res. 2022 Jan 23:103213.
  6. San Miguel-Ruiz JE, Polly D, Albersheim M, Sembrano J, Takahashi T, Lender P, Martin CT.Is the Implant in Bone? The Accuracy of CT and Fluoroscopic Imaging for Detecting Malpositioned Pelvic Screw and SI Fusion Implants. Iowa Orthop J. 2021;41(1):89-94.
  7. Quade J, Busel G, Beebe M, Auston D, Shah AR, Infante A, Maxson B, Watson D, Sanders RW, Mir HR.Symptomatic Iliosacral Screw Removal After Pelvic Trauma-Incidence and Clinical Impact. J Orthop Trauma. 2019 Jul;33(7):351-353.
  8. Berger-Groch J, Lueers M, Rueger JM, Lehmann W, Thiesen D, Kolb JP, Hartel MJ, Grossterlinden LG.Accuracy of navigated and conventional iliosacral screw placement in B- and C-type pelvic ring fractures. Eur J Trauma Emerg Surg. 2020 Feb;46(1):107-113.
  9. Maslow J, Collinge CA. Risks to the Superior Gluteal Neurovascular Bundle During Iliosacral and Transsacral Screw Fixation: A Computed Tomogram Arteriography Study. J Orthop Trauma. 2017 Dec;31(12):640-643.
  10. Araiza ET, Medda S, Plate JF, Marquez-Lara A, Trammell AP, Aran FS, Lara D, Danelson K, Halvorson JJ, Carroll EA, Pilson HT.Comparing the Efficiency, Radiation Exposure, and Accuracy Using C-Arm versus O-Arm With 3D Navigation in Placement of Transiliac-Transsacral and Iliosacral Screws: A Cadaveric Study Evaluating an Early Career Surgeon.

J Orthop Trauma. 2020 Jun;34(6):302-306.

We also added following references:

  1. Boudissa M, Carmagnac D, Kerschbaumer G, Ruatti S, Tonetti J. Screw Misplacement in Percutaneous Posterior Pelvic Iliosacral Screwing with and without Navigation: A Prospective Clinical Study of 174 Screws in 127 Patients. Orthop Traumatol Surg Res. 2022 Jan 23:103213.
  2. Tonetti J, Boudissa M, Kerschbaumer G, Seurat O. Role of 3D intraoperative imaging in orthopedic and trauma surgery. Orthop Traumatol Surg Res (2019);106(1S):S19-S25.
  3. Theologis AA, Burch S, Pekmezci M. Placement of iliosacral screw using 3D image-guided (O-Arm) technology and Stealth Navigation. Bone Joint J 2016;98-B:696-702
  4. Rommens PM, Nolte EM, Hopf J, Wagner D, Hofmann A, Hessmann M.Safety and efficacy of 2D-fluoroscopy-based iliosacral screw osteosynthesis: results of a retrospective monocentric study. Eur J Trauma Emerg Surg. 2021 Dec;47(6):1687-1698.
  5. Eastman JG, Routt Jr M. Correlating preoperative imaging with intraoperative fluoroscopy in iliosacral screw placement. J Orthopaed Traumatol 2015,16:309-316

We do hope that so far, we have improved the manuscript sufficiently to the major remarks. We are very glad for the reviewer's valuable comments.

Round 2

Reviewer 2 Report

Good revision; no deficiencies noted.